# Growth, Gas Exchange, and Boron Distribution Characteristics in Two Grape Species Plants under Boron Deficiency Condition

Rong Wei [1,2], Mei Huang [1,2], Dong Huang [1,2], Jinzhong Zhou [1,2], Xuejun Pan [1,2,*] and Wen'e Zhang [2,*]

1   Engineering Research Center for Fruit Crops of Guizhou Province, Guiyang 550025, China; gs.rwei20@gzu.edu.cn (R.W.); gs.zkhao20@gzu.edu.cn (M.H.); donghuang@gzu.edu.cn (D.H.); gs.hucai20@gzu.edu.cn (J.Z.)
2   College of Agriculture, Guizhou University, Guiyang 550025, China
*   Correspondence: xjpan@gzu.edu.cn (X.P.); agr.wezhang@gzu.edu.cn (W.Z.); Tel.: +86-187-8673-0619 (W.Z.)

**Abstract:** The boron (B) deficiency tolerance capacity of two grape materials, 'Xishui-4' (*Vitis flexuosa*) and 'Crystal' (*V. vinifera* × *V. labrusca*), were evaluated using a potted experiment in order to identify the B-use efficiency of grape and screen B-efficient grape resources. The sterile lines of two genotypes of grape were used as test materials, and a large number of test-tube seedlings were obtained through rapid propagation. The test-tube seedlings were acclimatization and transplanted, and the tested seedlings were treated with B stress after survival. In this experiment, the materials were cultured in nutrient solution, which contained 0.00 (B0), 0.25 (B1), and 0.50 (control) mg·L$^{-1}$ B concentrations, and the two genotypes of grape seedlings were cultured in vitro. The results were counted after 60 days of culture. The results showed that the B deficiency significantly reduced the growth parameters such as plant height, leaf area, total root length, and dry biomass of the two genotypes, and the inhibition of 'Crystal' growth parameters was greater than that of 'Xishui-4'. Moreover, the B deficiency also affected photosynthesis of the two genotypes, such as decreased leaf photosynthetic pigments, net photosynthesis rate, transpiration rate, stomatal conductance, intercellular carbon dioxide concentration, and stomatal density. Interestingly, the decrease ranges of 'Crystal' were greater than those of 'Xishui-4', indicating that 'Crystal' photosynthesis was more susceptible to B deficiency. Under the control condition, the concentration and accumulation of B in 'Crystal' were significantly higher than those in 'Xishui-4'. However, under the condition of B deficiency, the B concentration, accumulation amount, accumulation rate, utilization index, and tolerance index of 'Xishui-4' were higher than those of 'Crystal', and the B transport capacity of 'Xishui-4' was more stable, indicating that 'Xishui-4' had a better tolerance against B-deficient stress than 'Crystal' did. Therefore, 'Xishui-4' is a plant with strong adaptability to B deficiency stress, which can be used as B efficient grape resources and a genetic improvement of B efficient grape.

**Keywords:** boron deficiency; *Vitis*; growth; photosynthesis; tolerance



## 1. Introduction

As one of the essential trace elements, boron (B) plays an important function, such as participating in photosynthesis, nitrogen fixation, and respiratory metabolism [1,2], in the growth and development of higher plants [3,4]. Furthermore, B is involved in the formation of reproductive parts of plants, such as pollen tube growth, seed propagation, and carbohydrates transportation [5–7]. In addition, it plays a fundamental function in the structural and functional integrity of the cell wall and membranes, cell elongation and division, phenol transport and metabolism, ion fluxes across the membranes, etc. [8,9].

B deficiency is a widespread problem in agriculture at present. With the gradual reduction of the amount of micronutrients and the prevalence of B-deficient soils around the world, 132 crops in more than 80 countries have been severely reduced due to B deficiency alone [10,11], indicating that B deficiency is increasingly becoming an important

limiting factor in agricultural production [12]. The symptoms of B deficiency are varied and previous studies have shown that B deficiency damages various biochemical and physiological processes of crops, such as inhibiting apical growth of the shoots, curved leaves, repressed roots elongation, pollen reproductive abortion, and reduced flowering or early flowering, resulting in a continuous decline in output and quality [12,13].

Unlike other nutrients, the range between the deficiency and excess of B in soil is very narrow, and besides the soil itself, the comprehensive external environmental conditions are also very important in practical production, which makes it particularly difficult to manage B in agricultural management [14]. Therefore, one of the goals of sustainable agricultural systems is to develop and improve the B use efficiency (BUE) of plants (i.e., the ability of plants to absorb and utilize B for maximum yields). In general, plants preferentially distribute in environments where B concentrations are exactly within their normal range. In nature, however, plants are often found in soils where B is insufficient or too fertile. Therefore, in order to adapt to different environments, plants must optimize their absorption and distribution mechanisms of B to avoid B deficiency or toxicity [15]. Studies have found that a wide range of B-efficiency genotypes have been found in a variety of plants [16]. Therefore, it is speculated that different plant species or genotypes have different mechanisms and abilities to acquire, distribute, and utilize B in soil, which may lead to differences in B efficiency among them [17].

The B deficiency area of China's farming soil is up to 33 million hectares (hm$^2$) [18]. Guizhou province is located in Southwest China. As a typical karst mountainous area, the local soil erosion is serious and the B content in parent material is seriously lacking [19]. Guizhou has a complex terrain and diverse climate, and abundant grape plant resources [20]. The native soil of local wild grapes is generally B-deficient [20], but most grape plants grow well, indicating that the gene bank may contain grape species with high B efficiency. Pan et al. [20] found that the symptoms of *V. flexuosa* were slighter than those of other grapes under B-deficiency conditions. The result indicated that the wild grape species has developed a special adaptation mechanism to B deficiency concentration in long-term evolution. Huang et al. [18] evaluated the B efficiency of 7 different grape genotypes under B deficiency stress, and the results showed that the total B efficiency of the wild grape population native to the karst mountainous area of Guizhou province was higher than those of European grape and Euro-American hybrid. These results provided material support and a theoretical basis for screening B-efficient grape resources and the genetic improvement of B efficient grape.

However, until now, the response difference to B concentration between the wild grape species and the grape cultivar has been virtually nonexistent in the literature. Thus, understanding the mechanisms whereby B accumulates and transports in grape could provide a reference for the breeding of B-efficient grape varieties, as well as for the development of plants with increased BUE. The purposes of this study were (1) to assess the effects of B deficiency on plant growth, chlorophyll content, photosynthesis, stomatal characteristics, and B concentrations and distribution in wild species 'Xishui-4' (*V. flexuosa)*, which is originated in Guizhou province, and cultivar 'Crystal' (*V. vinifera* × *V. labrusca*); and (2) to further compare the tolerance of the two grapes species to different reactions of B deficiency according to the results of (1).

## 2. Materials and Methods

### 2.1. Plant Materials and Experimental Conditions

Plantlets of wild grape 'Xishui-4' (*V. flexuosa*) and cultivar 'Crystal' (*V. vinifera* × *V. labrusca*) were used as experimental materials, which were all taken from the grape planting resource nursery of The College of Agriculture, Guizhou University, and carried out in Guizhou Fruit Engineering Technology Research Center. According to the previous research [21,22], sterile lines of two genotypes of grapes were obtained by adjustment, and a large number of test-tube seedlings were obtained by rapid propagation. The rooting medium was 1/2MS + IAA 0.1 mg·L$^{-1}$ + IBA 0.1–0.15 mg·L$^{-1}$ + 6-BA 0.005 mg·L$^{-1}$. After about 45 d

of culture, all test-tube seedlings were transplanted and tested according to the method of Pan et al. [23]. After the tested seedlings survived, they were treated with B stress.

In this experiment, the material was cultured in nutrient solution. Previous studies found that the suitable concentration for grape plant growth was 0.50 mg·L$^{-1}$. Therefore, three B concentrations were selected in this study, which were 0.00 (B0), 0.25 (B1), and 0.50 (control) mg·L$^{-1}$ [24]. The nutrient solution consisted of 1/10 Hoagland and Arnon [25]. When the nutrient solution was configured, boron was added to the nutrient solution in the form of $H_3BO_3$ according to the concentration of B element, and the content of other elements remained unchanged (Table 1). All the drugs used in the preparation of the nutrient solution were extremely pure (GR: 99.8%), and all the water used in the test was ultra-pure water (UP) prepared by an AKSW-24 pure water meter, where the resistivity reached 18M $\omega$ *cm. The initial pH of the nutrient solution was 6.30, adjusted with 0.1 mol/L HCl or NaOH. The experiment consisted of 3 treatments with 3 replicates per treatment and 5 grape seedlings per replicate.

**Table 1.** The composition of nutrient solution (mg·L$^{-1}$).

| Boron Concentrations | Microelement Nutrition 1/10 Hoagland | | Microelement Nutrition Arnon | |
|---|---|---|---|---|
| | Chemicals | Contents | Chemicals | Contents |
| 0.00 (B0) | $KNO_3$ | 50.60 | $MnCl_2 \cdot 4H_2O$ | 1.81 |
| 0.25 (B1) | $NH_4NO_3$ | 8.00 | $CuSO_4 \cdot 5H_2O$ | 0.08 |
| 0.50 (control) | $KH_2PO_4$ | 13.60 | $ZnSO_4 \cdot 7H_2O$ | 0.22 |
| | $MgSO_4$ | 49.30 | $(NH_4)_6MoO_{24} \cdot 4H_2O$ | 0.02 |
| | $Ca(NO_3)_2 \cdot 4H_2O$ | 94.50 | $Na_2Fe$-EDTA | 4.20 |

Through the cultivation of transplanting and refining seedlings in the early stage, when the transplanting seedlings grew to 5–7 true leaves, seedlings with basically the same growth and size were selected, the perlite in their roots was washed with distilled water, and the seedlings were transplanted into a black plastic box (volume 13.5 L) and cultured with nutrient solution. The black plastic lid used in the test was perforated for planting material. Before use, the plastic box was soaked and scrubbed with 1:3 hot HCl to eliminate B pollution from the test. First, the material was cultured in water for 3 days, changed to 1/10 total nutrient solution for 4 d, and then cultured with different B concentrations of nutrient solution with a pH value of 6.30. During the culture period, ventilation was performed once a day, and the culture medium was replaced every 7 d with 12.00 L each time. The plastic box was rinsed with 1:40 HCl every 14 days. The experiment was carried out in an artificial climate chamber with the same culture environment at 25 ± 3 °C (relative air humidity, 55 ± 5%; illumination daily, 12 h (7:00–19:00); photosynthetic photon flux, 200 μmol·m$^{-2}$·s$^{-1}$).

### 2.2. Growth of Grape Seedlings Measurements

After 60 d of growth, each treatment randomly selected 3 grape seedlings with basically the same growth status to measure the height of the plant with a scale (accuracy to millimeters), and its average value was taken as the plant height of the treatment. The experimental materials were removed from the nutrient solution, washed with ultrapure water, and dried quickly with filter paper. They were divided into roots, stems, and leaves, and their fresh weight and dry weight were determined by an electronic balance. The leaf areas were measured using a leaf area meter (Li-3100A, LI-COR Biosciences Inc, Lincoln, NB, USA) (specific leaf area = leaf area /leaf dry weight). Total root length, total surface area, total volume, mean root diameter, and root tip number were measured by a Nuscan700 desktop root scanner. Each treatment was repeated 5 times, and the mean value was taken as the determination result.

*2.3. Determination of B Content and B-Related Parameters*

The experimental materials were removed from the nutrient solution and cleaned with ultrapure water, and the plant height, leaf area, and fresh weight of each part were determined. The plant height, leaf area, and fresh weight of each part were determined in an oven at 105 °C for 20 min, and then dried to a constant weight at 65 °C. After measuring the dry weight of the dried samples, they were crushed into powder by a plant mill, screened through 65 meshes, and stored in plastic bags for future use. The parameters of B, including concentration, accumulation, accumulation rate, distribution rate, transport factor (TF), utilization index, and tolerance index (TI), were determined.

The drying sample was weighed into a triangle flask, added with boron-free water to boil and dissolve, and centrifuged to separate the clear liquid after cooling (2 drops of $CaCl_2$ could be added to accelerate clarification). The clear liquid was taken into an evaporation dish, curcumin solution was added, and water bath evaporative drying to residual water was carried out, where this process showed red. The evaporated sample was dissolved in alcohol, filtered into a colorimetric tank, colorimetric at a 550 nm wavelength, and the B content was finally obtained [26].

B accumulation = $\sum$ (organ B content (ug/g) × organ dry weight) (g/plant) [27].

Total B accumulation rate = $\sum$ (organ B content × organ dry weight)/duration of B treatment × total dry weight [28].

Boron distribution rate = boron accumulation in organs/boron accumulation in whole plant [29].

TF = the concentration of B in the latter part/the concentration of B in the former part [30].

Boron utilization index = total dry biomass (mg)/total boron accumulation (ug) [31].

TI = biomass under low boron stress/biomass under normal boron supply ×100% [32].

*2.4. Leaf Photosynthetic Pigment Determination*

The fresh tissue from interveinal leaf-area was ground using a mortar and placed into a solution containing 80% (*v*/*v*) acetone. It was extracted in the dark at room temperature (10–30 °C). After the material was completely white, the absorbance of the liquid was measured with a spectrophotometer (Shimadzu UV-1700, Tokyo, Japan) at 664, 662, 644, and 440 nm. The chlorophyll content was determined and calculated by the formula of the acetone method [33,34].

$$Chla\ (mg{\cdot}g^{-1}FW) = (12.71\ D_{663} - 2.59\ D_{645}) \times [V/(1000 \times W)]$$

$$Chlb\ (mg{\cdot}g^{-1}FW) = (22.88\ D_{645} - 4.67\ D_{663}) \times [V/(1000 \times W)]$$

$$Car\ (mg{\cdot}g^{-1}FW) = (4.7 \times D_{440} - 0.27 \times Chl\ (a + b)) \times [V/(1000 \times W)]$$

*2.5. Leaf Gas Exchange Parameters*

When the three leaves of each plant were fully expanded, the plant representing the overall growth of each treatment was selected for listing and registration. The net photosynthesis rate (Pn), the transpiration rate (Tr), the stomatal conductance (Gs), and the intercellular carbon dioxide concentration (Ci) of the leaves were measured using a portable photosynthesis system (Li-6400; LiCor Inc., Lincoln, NB, USA) with an attached LED light source from 9:00 to 12:00 am on a clear day to determine the light and parameters of fully unfolded healthy leaves of different plants. Three strains were tested in each treatment for 3 replicates, and the parameters were: leaf temperature, 30 ± 2 °C; light intensity, 1000 μmol·m$^{-2}$·s$^{-1}$; relative humidity, 70%; $CO_2$ concentration of reference room, 450–480 μmol·mol$^{-1}$; gas flow, 450 mol·s$^{-1}$. The readings of each data point were recorded automatically after being stable for 2 min.

### 2.6. Leaf Stomata Observation

The following properties of the third mature leaf per plant were measured: stomatal length, stomatal width, and stomatal density. Measurement method: fresh leaves were taken and tweezers were used to directly tear the lower epidermis in the middle of the leaves (avoiding the main veins), a layer of gum cotton was brushed on both sides of the main veins, and then the film was removed with tweezers and put on the slide as a temporary slide. Twenty fields were randomly selected for each sample, the long and short axes of stomatal apparatus were measured, and the stomatal density (number/mm$^2$) was calculated by dividing the number of pores in each field by the field area. A temporary slide was observed with an OLYMpusBX40 digital microscope. The stomatal length and width were measured under high magnification ($10 \times 40$), and the stomatal density was measured under low magnification ($10 \times 10$).

### 2.7. Statistical Analysis

Excel and SPSS-17 (SPSS Inc., Chicago, IL, USA) were used for the statistical analysis. A one-way ANOVA was used to determine the significance of the differences ($p < 0.05$) among treatments, and Duncan's multiple range test was used to compare the data.

## 3. Results

### 3.1. Plant Growth

Exposing two grape genotypes to different levels of B solution resulted in reductions in growth, as shown in Figure 1 and Table 2. Compared with the control, B deficiency plants had short shoots, yellowing leaves, and short roots, indicating that B deficiency significantly inhibited the growth of grape seedlings of the two genotypes. After treatment for 60 d, old leaves of 'Xishui-4' were slightly yellowed at 0.00 mg·L$^{-1}$ (Figure 1a). However, the yellow fraction spots were observed in old leaves of 'Crystal' under deficient boron (0.00 and 0.25 mg·L$^{-1}$), and gradually expanded to the middle leaves (Figure 1b), which leads to the conclusion that under B deficiency stress, the leaf yellowing rate of 'Crystal' was more evident than that of 'Xishui-4'.

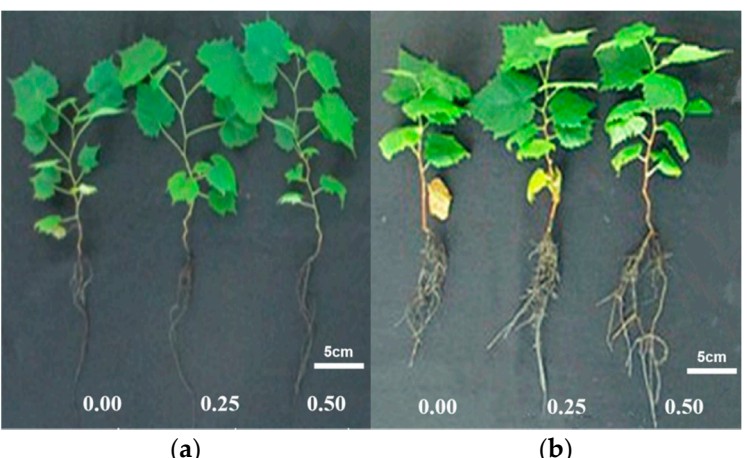

        (**a**)               (**b**)

**Figure 1.** Growth status of two genotypes treated with 0.00, 0.25, and 0.50 mg·L$^{-1}$ B solution for 60 days: (**a**) 'Xishui-4'; (**b**) 'Crystal'.

As shown in Table 2, seedlings of two grape genotypes treated with B deficiency for 60 days had lower levels of biomass than control plants. Furthermore, when 0.00 mg·L$^{-1}$ B was added, the plant height, leaf area, total root length, and dry biomass of the two genotypes were significantly decreased compared with the control. The results showed that B deficiency inhibited the growth of two genotypes to a certain extent. However, the growth parameters of 'Xishui-4' were significantly higher than those of 'Crystal' under both control and B deficiency conditions. Compared with the control (0.05 mg·L$^{-1}$ B), the plant height, leaf area, total root length, and dry biomass of 'Xishui-4' without B nutrient

solution (0.00 mg·L$^{-1}$) significantly decreased by 16.83%, 20.11%, 14.02%, and 9.00%, and those of 'Crystal' decreased by 23.81%, 28.37%, 17.49%, and 17.71%, respectively. These results indicated that the B deficiency inhibited the growth parameters of 'Crystal' more obviously than that of 'Xishui-4'.

**Table 2.** Effects of B-deficiency on the growth of two grape seedlings.

| Genotypes | B Concentrations/ (mg·L$^{-1}$) | Plant Height/(cm) | Leaf Area/(cm$^2$) | Total Root Length/(cm) | Root Number | Dry Biomass/ (g)/DW |
|---|---|---|---|---|---|---|
| 'Xishui-4' | 0.00 (B0) | 17.94 ± 1.30c * | 16.45 ± 1.03c * | 191.64 ± 8.18b * | 372.17 ± 36.38a * | 0.354 ± 0.007b * |
| | 0.25 (B1) | 19.41 ± 0.70b * | 18.53 ± 0.56b * | 203.27 ± 12.41b * | 396.00 ± 33.95a * | 0.379 ± 0.007a * |
| | 0.50 (control) | 21.57 ± 1.26a * | 20.59 ± 0.86a * | 222.90 ± 10.76a * | 417.67 ± 35.67a * | 0.389 ± 0.019a * |
| 'Crystal' | 0.00 (B0) | 13.89 ± 0.86c | 9.24 ± 0.61c | 156.31 ± 7.99c | 352.17 ± 32.85a | 0.302 ± 0.006c |
| | 0.25 (B1) | 16.33 ± 0.81b | 10.70 ± 0.31b | 170.21 ± 6.46b | 381.17 ± 34.17a | 0.332 ± 0.015b |
| | 0.50 (control) | 18.23 ± 1.30a | 12.90 ± 0.64a | 189.45 ± 9.63a | 394.17 ± 32.97a | 0.367 ± 0.015a |

Note: Data are shown as the mean ± S.E. Different lowercase letters in the table indicate significant differences between the treatments ($p < 0.05$). * represents significant differences between the two genotypes ($p < 0.05$).

### 3.2. Determination of B Content and B-Related Parameters

Compared with the control, two grape genotypes treated with 0.00 and 0.25 mg·L$^{-1}$ B for 60 d significantly decreased the B concentrations in both shoots and roots ($p < 0.05$) (Figure 2). When grapes were not supplied with B, shoot and root B concentrations of 'Xishui-4' were reduced by 32.42% and 17.93%, and those of 'Crystal' were reduced by 43.58% and 37.30%, respectively. In addition, higher B concentration values were recorded in the root of 'Xishui-4' than in 'Crystal' (Figure 2b), and the shoot was contrasted to root (Figure 2a). In general, the shoots represented the dominant sites, regardless of the genotypes and B treatments. Under the control condition, the concentration of B in 'Crystal' (111 ± 0.61a mg·kg$^{-1}$ DW) was significantly higher than that of 'Xishui-4' (97.5 ± 1.30b mg·kg$^{-1}$ DW). Therefore, the B concentration of 'Crystal' was significantly higher than that of 'Xishui-4' under the control condition, indicating that 'Crystal' needed more B than 'Xishui-4'.

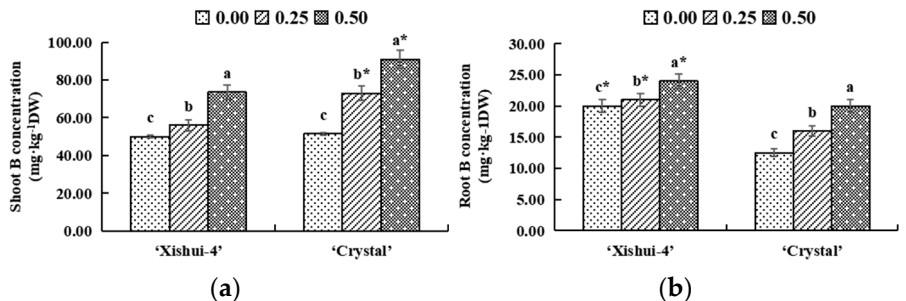

**Figure 2.** Effects of B-deficiency on B concentration of two grape genotypes: (**a**) shoot; (**b**) root. Different lowercase letters within each column are significantly different at $p < 0.05$ between treatments. * is significantly different at $p < 0.05$ between genotypes.

The B accumulation in two genotypes plants was similar to that of B concentration (Figure 3). When treated with different B concentrations, the B accumulation in the shoot of two genotypes was higher than that of the root. B deficiency also significantly reduced the B accumulation in different organs of two genotypes, and the reductions of 'Crystal' were higher than those of 'Xishui-4'. Under the control, the B accumulation in the shoot and root of 'Xishui-4' was significantly lower than that of 'Crystal', but it was the opposite with the nutrient solution without B (0.00 mg·L$^{-1}$), indicating that 'Crystal' used more B under the condition of B deficiency than 'Xishui-4' did.

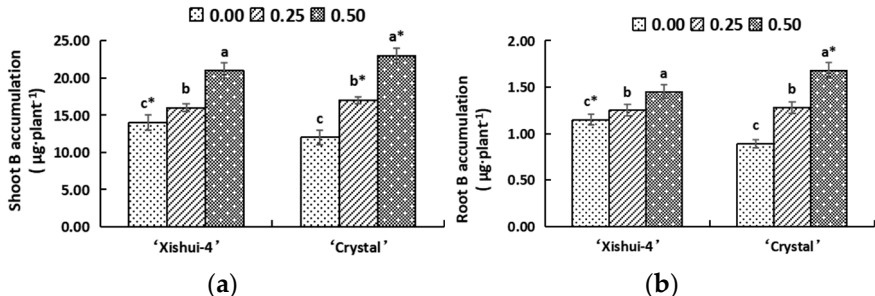

**Figure 3.** Effects of B-deficiency on B accumulation (µg·plant$^{-1}$) of two grape genotypes: (**a**) shoot; (**b**) root. Different lowercase letters within each column are significantly different at $p < 0.05$ between treatments. * is significantly different at $p < 0.05$ between genotypes.

The total B accumulation and accumulation rate of two grape genotypes were significantly influenced by the decrease in B concentrations (Figure 4). Under the control condition, the accumulation of B in 'Crystal' (27.5 ± 0.56a µg·plant$^{-1}$) was significantly higher than that of 'Xishui-4' (25 ± 0.81b µg·plant$^{-1}$). When the B concentration decreased from 0.50 to 0.00 mg·L$^{-1}$, the total B accumulation of the two genotypes decreased significantly, and the decrease extent of 'Crystal' (49.45%) was more than that of 'Xishui-4' (36.00%) (Figure 4a). The total B accumulation rate variation was in agreement with the change in B accumulation (Figure 4b). With the nutrient solution without B (0.00 mg·L$^{-1}$), the total B accumulation rate of 'Xishui-4' was reduced by 41.76%, and that of 'Crystal' was reduced by 58.41%. These results indicated that 'Xishui-4' had a smaller reduction in B accumulation rate caused by B deficiency, and the B accumulation in plants was more stable.

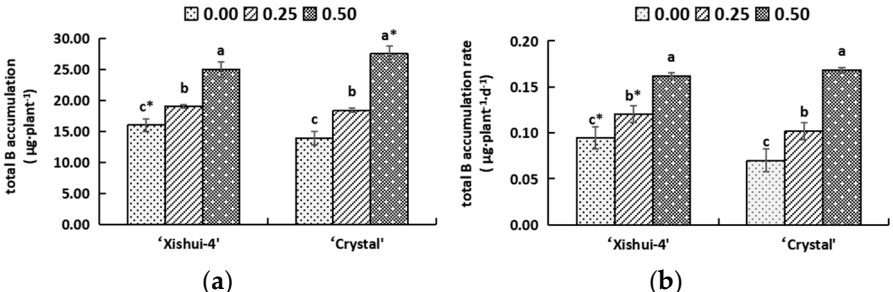

**Figure 4.** Effects of B-deficiency on B accumulation of two grape genotypes: (**a**) total B accumulation; (**b**) total B accumulation rate. Different lowercase letters within each column are significantly different at $p < 0.05$ between treatments. * is significantly different at $p < 0.05$ between genotypes.

Figure 5 shows the distribution proportion of B content in the two genotypes under the treatment of nutrient solution with different B concentrations. According to the proportion of B content in the shoot and root, it was found that the B content in the shoot was higher than that in the root of both genotypes regardless of B level. When B decreased from 0.25 to 0.00 mg·L$^{-1}$, the fractions of B distributed to shoots were less in 'Xishui-4' (93.52–92.79%) than in 'Crystal' (93.75–93.21%), indicating that the distribution and transport of 'Crystal' from the shoot to leaf was more inhibited than those of 'Xishui-4'.

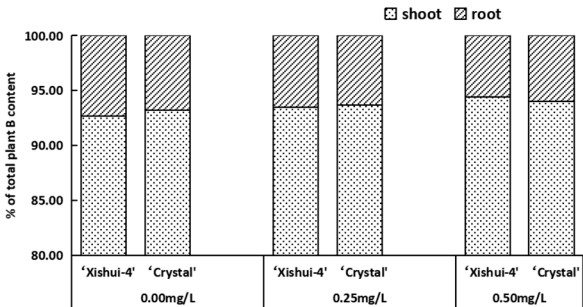

**Figure 5.** Effects of B-deficiency on B distribution proportion of two grape genotypes.

Figure 6 shows that the transport factor of 'Xishui-4' was significantly lower than that of 'Crystal'. It was found that there was no significant difference in the transport factor (root→stem) of 'Xishui-4' under the three different B concentrations, while the transport factor (root→stem) of 'Crystal' increased with the decrease in the B concentration of nutrient solution (Figure 6a). The result showed that the B content in the roots and stems of 'Xishui-4' did not change significantly with the decrease in the B concentration of nutrient solution. However, a large amount of B in the 'Crystal' root flowed to the stem, leading to a gradual decrease in the concentration of B in the root, and a gradual increase in the concentration of B in the stem at the same time. Figure 6b shows that with the decrease in the B concentration of nutrient solution, the transport factor (stem→leaf) of two grape genotypes began to decline, indicating that when the B deficiency stress became more serious, the concentration of B flowing from the stem to the leaf was blocked. The two transport factors and their variation range of 'Xishui-4' were smaller than those of 'Crystal', indicating that 'Xishui-4' had better adaptability to low B stress than 'Crystal'.

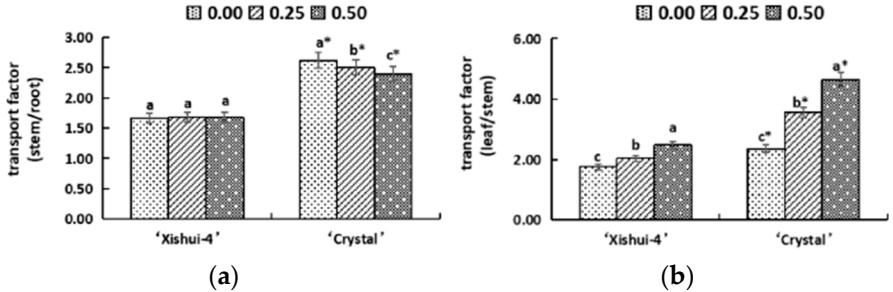

(a)                                                     (b)

**Figure 6.** Effects of B-deficiency on transport factor of two grape genotypes: (**a**) TF $_{root→stem}$; (**b**) TF $_{steam→leaf}$. Different lowercase letters within each column are significantly different at $p < 0.05$ between treatments. * is significantly different at $p < 0.05$ between genotypes.

To evaluate the capability of grape to adapt and tolerate B deficiency, the B utilization index and tolerance index were calculated (Figure 7). According to Figure 7a, under low B stress, the B utilization index of both genotypes was significantly higher than that of the control, indicating that the B utilization efficiency of the two genotypes was significantly improved under the condition of B deficiency. As the B supply decreased from 0.50 to 0.00 mg·L$^{-1}$, the B utilization index of 'Xishui-4' was significantly higher than that of 'Crystal' by 14.20%, 9.57%, and 1.81%.

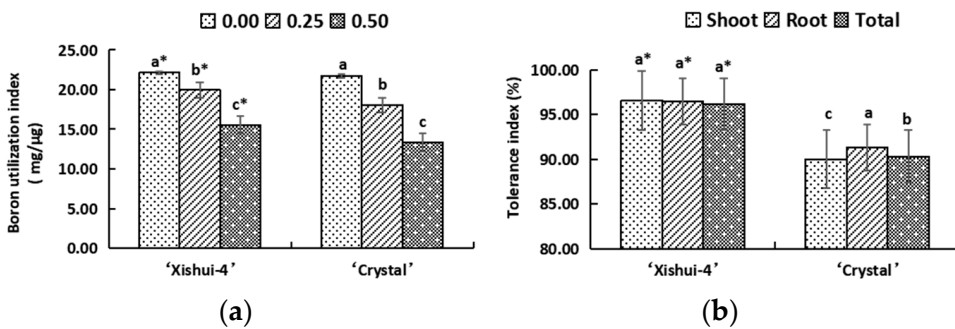

**Figure 7.** Effects of B-deficiency on adaptability difference of two grape genotypes: (**a**) utilization index; (**b**) tolerance index. Different lowercase letters within each column are significantly different at $p < 0.05$ between treatments. * is significantly different at $p < 0.05$ between genotypes.

### 3.3. Leaf Photosynthetic Pigment

Table 3 shows the effects of B deficiency on chlorophyll parameters of the two genotypes. We found that the content of pigments (Chl. a, Chl. b, Car., and Chl. A + b) of the two grape genotypes decreased gradually with the decrease in concentrations of B. Compared with its control (0.50 mg·L$^{-1}$ B), the contents of Chl. a, Chl. b, Car., and Chl. (a + b) of 'Xishui-4' at the 0.00 mg·L$^{-1}$ B treatment significantly decreased by 16.83%, 20.10%, 14.02%, and 10.89%, respectively. However, those of 'Crystal' decreased by 23.80%, 28.37%, 17.49%, and 10.66%, respectively.

**Table 3.** Effects of B-deficiency on the leaf photosynthetic pigments of two grape genotypes.

| Genotypes | B Concentrations/ (mg·L$^{-1}$) | Chl. a/ (mg·g$^{-1}$ FW) | Chl. b/ (mg·g$^{-1}$ FW) | Car/(mg·g$^{-1}$ FW) | Chl. (a + b)/ (mg·g$^{-1}$ FW) | Chl. (a/b) |
|---|---|---|---|---|---|---|
| 'Xishui-4' | 0.00 (B0) | 17.94 ± 1.30c * | 16.45 ± 1.03c * | 191.64 ± 8.18b * | 372.17 ± 36.38a * | 0.354 ± 0.007b * |
| | 0.25 (B1) | 19.41 ± 0.70b * | 18.53 ± 0.56b * | 203.27 ± 12.41b * | 396.00 ± 33.95a * | 0.379 ± 0.007a * |
| | 0.50 (control) | 21.57 ± 1.26a * | 20.59 ± 0.86a * | 222.90 ± 10.76a * | 417.67 ± 35.67a * | 0.389 ± 0.019a * |
| 'Crystal' | 0.00 (B0) | 13.89 ± 0.86c | 9.24 ± 0.61c | 156.31 ± 7.99c | 352.17 ± 32.85a | 0.302 ± 0.006c |
| | 0.25 (B1) | 16.33 ± 0.81b | 10.70 ± 0.31b | 170.21 ± 6.46b | 381.17 ± 34.17a | 0.332 ± 0.015b |
| | 0.00 (B0) | 18.23 ± 1.30a | 12.90 ± 0.64a | 189.45 ± 9.63a | 394.17 ± 32.97a | 0.367 ± 0.015a |

Note: Data are shown as the mean ± S.E. Different lowercase letters in the table indicate significant differences between the treatments ($p < 0.05$). * represents significant differences between the two genotypes ($p < 0.05$).

### 3.4. Gas Exchange Parameters

Different concentrations of B treatment had a great influence on the leaf gas exchange parameters of the two grape genotypes. We observed that the B deficiency stress inhibited leaf photosynthesis (Table 4), and lower values were recorded in 'Xishui-4' than in 'Crystal'. When the plants were treated with 0.00 mg·L$^{-1}$, the Pn, Tr, Gs, and Ci of 'Xishui-4' significantly decreased by 36.94%, 15.15%, 17.54%, and 9.09%, and those of 'Crystal' decreased by 38.70%, 15.57%, 18.26%, and 10.51%, respectively, compared with their control (0.50 mg·L$^{-1}$) treatment. The results showed that under the B deficiency condition, the inhibition degree of gas exchange of 'Xishui-4' was lower than that of 'Crystal'.

**Table 4.** Effects of B-deficiency on the photosynthesis of two grape genotypes.

| Genotypes | B Concentrations/ (mg·L$^{-1}$) | Net Photosynthesis Rate (Pn)/(µmol·m$^{-2}$·s$^{-1}$) | Transpiration Rate (Tr)/(mmol·m$^{-2}$·s$^{-1}$) | Stomatal Conductance (Gs)/(mol·m$^{-2}$·s$^{-1}$) | Intercellular Carbon Dioxide Concentration (Ci)/(µmol·mol$^{-1}$) |
|---|---|---|---|---|---|
| 'Xishui-4' | 0.00 (B0) | 0.70 ± 0.01c | 0.28 ± 0.01b | 9.40 ± 0.50b | 273.33 ± 1.53c |
| | 0.25 (B1) | 1.02 ± 0.04b | 0.31 ± 0.01a | 10.90 ± 0.60a | 285.67 ± 5.96b |
| | 0.50 (control) | 1.11 ± 0.01a | 0.33 ± 0.02a | 11.40 ± 0.40a | 300.67 ± 5.03a |
| 'Crystal' | 0.00 (B0) | 1.98 ± 0.11c * | 0.61 ± 0.01c * | 16.43 ± 0.60b * | 505.33 ± 2.52c * |
| | 0.25 (B1) | 2.72 ± 0.13b * | 0.65 ± 0.01b * | 19.40 ± 0.80a * | 539.00 ± 4.44b * |
| | 0.50 (control) | 3.23 ± 0.21a * | 0.74 ± 0.03a * | 20.10 ± 0.50a * | 564.67 ± 7.01a * |

Note: Data are shown as the mean ± S.E. Different lowercase letters in the table indicate significant differences between the treatments ($p < 0.05$). * represents significant differences between the two genotypes ($p < 0.05$).

### 3.5. Leaf Stomata Characteristic Observation

Table 5 shows the effects of B deficiency on stomatal length, width, and density of the two grape genotypes. The stomatal density and stomatal opening degree of the two genotypes also decreased progressively with the decrease in B concentrations in both grapes, but only the stomatal density decreased significantly. The stomatal density of the 'Xishui-4' and 'Crystal' were significantly reduced by 45.81% and 46.28% at the 0.00 mg·L$^{-1}$ treatment, when compared with their control treatment. The result showed that the reduction in stomatal density of 'Crystal' was higher than that of 'Xishui-4', indicating that the stomata characteristic of 'Crystal' was more susceptible to B deficiency than 'Xishui-4'.

**Table 5.** Effects of B-deficiency on the stomata characteristic of two grape seedlings.

| Genotypes | B Concentrations/ (mg·L$^{-1}$) | Stomatal Length/ µm | Stomatal Width/ µm | Stomatal Density/ Number·mm$^2$ |
|---|---|---|---|---|
| 'Xishui-4' | 0.00 (B0) | 7.16 ± 1.25a | 4.94 ± 0.81a | 50.03 ± 3.15c |
| | 0.25 (B1) | 7.66 ± 1.61a | 5.19 ± 0.79a | 84.81 ± 3.91b * |
| | 0.50 (control) | 7.78 ± 1.84a | 5.43 ± 0.67a | 92.32 ± 2.56a |
| 'Crystal' | 0.00 (B0) | 9.63 ± 0.56a * | 5.93 ± 0.56a * | 55.20 ± 6.77c * |
| | 0.25 (B1) | 9.51 ± 0.98a * | 5.93 ± 0.79a * | 78.98 ± 11.05b |
| | 0.50 (control) | 9.75 ± 0.93a * | 5.80 ± 1.08a * | 102.76 ± 10.89a * |

Note: Data are shown as the mean ± S.E. Different lowercase letters in the table indicate significant differences between the treatments ($p < 0.05$). * represents significant differences between the two genotypes ($p < 0.05$).

### 4. Discussion

In this research, we discussed the differential responses of plant growth, B content, B-related parameters, pigments content, photosynthesis, and stomata characteristics in two grape genotypes to different concentrations of B. In previous studies, B deficiency had a certain degree of influence on the growth of higher plants, such as inhibiting plant biomass accumulation, plant height, and root development [35]. In our experiment, it was found that B deficiency resulted in chlorosis, shorter roots, and stunted plants in both genotypes (Figure 1). Through the analysis of the growth parameters, we found that B deficiency significantly reduced plant height, leaf area, total root length, and dry biomass (Table 2), and this situation also existed in other plants [30,36]. Root elongation rate mainly depends on the elongation of root cells and the cell wall. Therefore, it is speculated that one of the reasons for the inhibition of root elongation may be that B deficiency caused the thickening of the root tip cell wall, which changed its structure, and then reduced the toughness of the cell wall, so that the new meristem cell of the root tip cannot complete the normal enlargement and division, finally leading to the root growth obstruction [30,36]. In addition, Dong [30] analyzed the relationship between the amino acid metabolism profile induced and changes in the shikimic acid pathway, and found that B deficiency can cause the accumulation of auxin in roots through the level of tryptophan in the shikimic acid pathway, which leads to the stagnation of root elongation, providing strong new evidence that B deficiency inhibits root elongation. B deficiency significantly reduced the

root length of cotton and rape, and inhibited the root length of sensitive cultivars more than that of insensitive cultivars [37]. In this study, it was also observed that the B deficiency inhibition degree on root parameters of 'Crystal' was greater than that of 'Xishui-4' (Table 2), indicating that the root of 'Crystal' was more sensitive than that of 'Xishui-4'. The results of other plants roots subjected to B deficiency stress were consistent with the results of this study [38,39].

The B efficiency of different genotypes is expressed in absorption and utilization. Boaretto et al. [40] observed different B requirements on oranges grafted on different rootstocks and pointed out that those B-sensitive cultivars had higher B requirements. Liu et al. [41] also observed similar results. In this study, it was found that the B concentration of the 'Crystal' was significantly higher than that of 'Xishui-4' under the control condition, while the B concentration and accumulation of 'Crystal' decreased at a faster rate than 'Xishui-4' in the deficiency of B (Figures 2 and 3), indicating that 'Crystal' needed more B and consumed more B than 'Xishui-4' under B deficiency stress. Under B deficiency stress, the B uptake of different cultivars decreased to different degrees, but the decrease degree of insensitive cultivars was less than that of sensitive cultivars. The B isotope test showed that the B absorption capacity of insensitive cultivars was stronger than sensitive cultivars under B deficiency stress [42]. This is consistent with Figure 4. Under the condition of B deficiency, the B accumulation amount and accumulation rate of 'Xishui-4' were significantly higher than those of 'Crystal', indicating that 'Xishui-4' had a stronger absorption capacity of B than 'Crystal'. The decrease in B accumulation rate was less than that of 'Crystal', indicating that the decrease degree was small. The B utilization capacity of plants generally includes a B requirement and distribution capacity. If plants have a lower B requirement, produces more biomass, and preferentially allocates B to other organs, the plant will be more tolerant to B deficiency stress. The B content varies greatly among different plants, which can reflect the B requirement of plants. The mobility of B in the phloem has been demonstrated by many studies; it was also found that there were significant differences in B mobility in broccoli and peanut with different B efficiencies, and B-insensitive cultivars showed a stronger redistribution ability than sensitive cultivars did [43,44]. As an important channel for B transport from the root to leaf, the shoot is responsible for regulating the distribution and transport of B in the root from the stem to the leaf when B is lacking, and for the normal growth of plants. By analyzing the percentage of B concentration in each part of two genotypes in the total B concentration of the whole plant, we found that B deficiency significantly reduced the percentage of B concentration in the shoot, but increased the percentage in the root. In the deficiency of B, the distribution capacity of B to leaves in the shoots of 'Crystal' was lower than that in 'Xishui-4', and the change range of transport factors was larger than that in the 'Xishui-4', indicating that 'Crystal' had a lower adaptability to low B stress than 'Xishui-4' did (Figures 5 and 6). In addition, according to Figure 7, these results indicated that 'Xishui-4' had a stronger B utilization ability than 'Crystal', and can produce more biomass per unit B supply under the condition of B deficiency. The results of tolerance index were consistent with the B utilization index, and each part of 'Xishui-4' was significantly higher than that of 'Crystal' (Figure 7b), indicating that the tolerance of 'Xishui-4' to B deficiency stress was better than that of 'Crystal'. In this experiment, the shoot distribution rate of B content and the transport factor of each organ of 'Crystal' was significantly higher than that of 'Xishui-4' under the B deficiency condition, which might be related to the higher B requirement and B transport capacity (Figures 5 and 6). However, with the change in external B supplement, there was no significant difference in the transport factor of 'Xishui-4' from the root to shoot, and its transport factors of each organ were smaller than those of 'Crystal', indicating that the B transport capacity of 'Xishui-4' was more stable than that of 'Crystal' (Figure 6). Therefore, 'Xishui-4' had a higher B utilization efficiency than 'Crystal'.

Leaf is an important nutrient organ for photosynthesis and food production. B deficiency generally leads to symptoms such as thick and brittle leaves, malformed curls, and rupture of main veins [45]. Oliveira et al. [46] conducted an anatomical study on cotton and

found that the differentiation of leaf phloem was inhibited by B deficiency. Liu et al. [47] found that the B deficiency significantly increased the proportion of sponge tissue and palisade tissue in citrus leaves through morphology and tissue structure observation, and sponge tissue cells showed an abnormal enlargement and increase, resulting in irregular thickening of the leaves. In addition, other scholars also observed that under B deficiency stress, leaves appeared yellowing and dieback, thickening and brittle, and veins swelling and rupture, etc. [48], and they speculated that B deficiency might cause the enlargement of spongy tissue cells in leaves, the increase in vascular bundle area and leaf thickness, and thus the tearing of the upper epidermis enclosing the vascular bundle [49]. However, in our study, it was found that there were no other obvious symptoms of B deficiency except yellowing in the leaves. It was speculated that the short period of B deficiency did not cause an abnormal increase in the area of vascular bundle tissue, and the changes in the internal structure of the plant were not enough to damage the epidermis structure. Table 3 indicates that under low B conditions, the synthesis capacity of photosynthetic pigments in leaves of 'Xishui-4' was stronger than that of 'Crystal', and the consumption capacity was lower than that of 'Crystal'. According to the previous research progress, the further analysis of tables (Tables 2–5) showed that B deficiency reduced chloroplast content and stomatal density of the two genotypes of grapes. It was speculated that this might lead to a decrease in fixation ability of plants to carbon dioxide, which led to a continuous decrease in intercellular carbon dioxide concentration, and thus affected the synthesis of photosynthetic products. Ultimately, it hindered the normal growth of the two genotypes of grapes. In addition, it was reported that some of the photosynthesis activities and photosynthetic capacity of citrus leaves were inhibited by B deficiency, while the activities of antioxidant enzymes showed a trend of upregulation. Due to B deficiency, photo-contracted compounds are blocked from entering glycolysis and pentose phosphate pathways, which leads to the excessive accumulation of carbohydrates, which are then converted into starch in chloroplasts and stored in chloroplasts. As a result, the utilization of sugars in leaves is blocked and the energy produced is reduced, leading to changes in photosynthesis and thus inhibiting plant growth [30,50,51]. Therefore, although many visible symptoms of B deficiency were not observed in the leaves, the internal structure of the leaves was altered to the extent that the normal photosynthesis was affected and the normal growth of the plants was hindered.

## 5. Conclusions

In this study, we found that the growth status of 'Xishui-4' (*V. flexuosa*) was better than that of 'Crystal' (*V. vinifera* × *V. labrusca*) under B-deficiency conditions, because the absorption and transportation of 'Xishui-4' were stronger than those of 'Crystal' under lower-concentration B conditions, which led to the chlorophyll synthesis ability being higher and the decline in photosynthesis of 'Xishui-4' being less than those of 'Crystal' when the B supply was deficient. Therefore, the leaf yellowing and the biomass reduction of 'Xishui-4' were lighter and smaller than those of 'Crystal'. This ability of 'Xishui-4' may be related to the long-term adaptation to a low-boron environment. Our study results indicated that the 'Xishui-4' was a special grape germplasm resource with a highly efficient utilization of B and could be used as a genetic improvement of B-efficient grape.

**Author Contributions:** All authors contributed to the study conception. Material preparation was performed by J.Z. Data collection was accomplished by M.H. Data analysis, experiment design, and supervision were performed by X.P., W.Z. and D.H. The draft of the manuscript was written by R.W. All authors commented on previous versions of the manuscript. All authors have read and agreed to the published version of the manuscript.

**Funding:** This research was funded by the National Natural Science Foundation of China, grant number 31560546, and the Hundred Levels Talent Training of Guizhou Province, grant number [2016]4038.

**Institutional Review Board Statement:** The study did not involve humans or animals.

**Informed Consent Statement:** The study did not involve humans.

**Data Availability Statement:** The data are available upon request from the corresponding author.

**Acknowledgments:** The authors thank Weiguo Fan for technical advice.

**Conflicts of Interest:** The authors declare no conflict of interest.

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
