# Peer review of "Growth, Gas Exchange, and Boron Distribution Characteristics in Two Grape Species Plants under Boron Deficiency Condition"

_horticulturae, doi:10.3390/horticulturae8050374_

Round 1

Reviewer 1 Report

The manuscript “Growth, gas exchange and boron distribution responses of two grape species to boron deficiency” is an interesting contribution. However, the statistical design of the research is not understood. In addition, I think that the authors should remade the analysis of variables and to perform a multivariate analysis accounting genotype and dosage of B application factors. Material and method sections should be better described. The writing is very messy, there are paragraphs that are not justified and others that are. There are several acronyms that are not mentioned in the first place and the tables are not self-explanatory. The writing is very poorly written, and the English is not very good. Conclusions should be remade according to the results and exposed data.

Tittle

L2-3: Where boron distribution was evaluated? This should be added in the tittle. In addition, grape species should be edited if it corresponds to grapevine or table grape.

Abstract

L9-11: Little information about material and methods should be added.

L16-17: Sentences cannot start with "and". Please remade this sentence for a better understanding.

L18: These control conditions should be explained before results.

L19: Numerical data should be added to know the difference in boron concentration.

Introduction

L83: Please, add a space in "x" to separate the species.

Material and methods

L86-164: The experiment should be explained much better, in the results it is spoken of 60 days after the treatments and in this section, they are not explained. The experimental design is very poorly explained and does not explain if there are block designs and repetitions for treatments.

L89: In think that the authors should have uniformity in the expression of the species and edit it along text.

L96: Please, add the highest or the lowest percentage of purity.

L99-104: This information should be shortened and only mention the treatments including in the final, the corresponding citation.

L105: What's means rooted plantlets? Is that the vines were grafted?

L109-111: The experimental design must be clearly explained

L120-136: This section should be better explained.

L138-139: Please, add the methodology in which was determined chlorophyll content.

L141-142: This information should be described above.

L144-151: Information about the protocol of measurement in terms of the properties of leaves should be added.

L154-156: Please, explain better this sentence.

L161-164: The authors should improve statistical analysis section adding the variables in which was determined ANOVA.

L162-164: The authors should perform a multifactorial analysis since there is two factors, variety and dosage of boron.

Results

L167-168: The authors should add "and Table 1" in the final step of the sentence. In this way, the differences should be numerically evidenced as was defined in the statistical analysis section.

L168-175: This paragraph is not justified. In addition, replace "serious" by "evidenced".

L176: These are not growth conditions. Please, add the correct term for this picture.

L179: This treatment (0.00 mg/L B) could be defined as control.

L181: Really, I don't understand the difference between apply 0.00 mg L B with the control.

L184: I don't understand why control is the application of 0.05 mg/L B.

L185: Leaves area is not a correct term, please, edit this word.

L194-198: This sentence is not well explained. Please, remade for a better understanding.

L204: In Figure 2 to 4, I think that the authors should add a uppercase letter to show differences between varieties.

L215-223: This paragraph is not justified.

L226-236: This paragraph is not justified

L228-233: This sentence should be placed in the discussion section.

L240-253: This paragraph is not justified.

L255: In Figure 6 to 7, I think that the authors should add an uppercase letter to show differences between varieties.

L258-261: How did you compare the genotypes with the control? For these reasons, it could be better to perform a MANOVA rather than ANOVA.

L261-263: This sentence is difficult to follow because you mention two dosage and three percentages. Please, be uniform.

L263-268: This information should be placed in the discussion section.

L272-273: This is not correct, please, make sure that the differences are statistical and according to what is shown in the Table.

L276-278: This sentence should be placed in the discussion section.

L280: Please, replace "Leaf" by "leaf"

L285-287: Please, explain the acronyms when are firstly mentioned.

L294: Tables should be self-described and explain acronyms.

L295: Please, describe better the tittle of the section.

Conclusions

L403-404: These sentences do not correspond to a scientific article, please, edit as corresponds.

Author Response

Dear Reviewer:

Thank you for giving us the opportunity to submit a revised draft of the manuscript “Growth, gas exchange and boron distribution characteristic in two grape species plant under boron deficiency condition” (Manuscript ID: horticulturae-1671817) for publication in the Journal of Horticulturae. We appreciate the time and effort that you dedicated to providing feedback on our manuscript and we are grateful for the insightful comments on and valuable improvements to our paper. We have incorporated the suggestions. As you are concerned, there are several problems that need to be addressed. Thus, according to the nice suggestions, we have made extensive corrections to our previous draft, the detailed corrections are listed below. All the changes are marked in red in the manuscript. Please see the report, in blue and highlighted yellow, for a point-by-point response to the comments and concerns.

Reviewer 2 Report

Dear Authors

The article is well written and contains an interesting experiment. However, I have some issues that need to be clarified before publication.

Here are some comments on the article:

Lines 184: Ë®...Compared with the control (0.05 mg·L-1 B) ” . Here I consider that 0.50 mg·L-1 B is correct variant. Please check and change.

Line 190: Table 1. I would suggest to identify more clearly in the table which is the control variant (e.g. 0.50 control) or another way to highlight the control. Is better for readers.

Line 280:   Table 2. Carotenoids ???.

In subchapter 2.4. Leaf Photosynthetic Pigment Determination you only mentioned and described the method for determining of chlorophyll. The method of determining the total or individual carotenoids content has not been described. Please check.

At the Table 2, but also tables 3 and 4 missing the explanation below the table that refers to the statistical calculation. You wrote in line 192 The same as below but I think it is appropriate to write under each table.

Author Response

(The authors gave the same response as above.)

Reviewer 3 Report

I think the article is ok, but only two cultivars are included in this so in the caser of grapevine this is not enough. Experiment is designed with three B concentrations , I think it should be at least four.

The observed parameters are well explained and presented. 

This article can be useful for China.

I recommend it for publication.

Author Response

(The authors gave the same response as above.)

Round 2

Reviewer 1 Report

The authors have taken into account the suggestions performed by the reviewer and the manuscript has improved considerably from its initial version.

This manuscript is a resubmission of an earlier submission. The following is a list of the peer review reports and author responses from that submission.